# A Micro Air Velocity Sensor for Measuring the Internal Environment of the Cold Air Ducts of Heating, Ventilation, and Air Conditioning Systems

**DOI:** 10.3390/mi13122198

**Published:** 2022-12-11

**Authors:** Chi-Yuan Lee, Xin-Wen Wang, Chen-Kai Liu

**Affiliations:** Department of Mechanical Engineering, Yuan Ze Fuel Cell Center, Yuan Ze University, Taoyuan 32003, Taiwan

**Keywords:** micro air velocity sensor, heating, ventilation and air conditioning

## Abstract

A wireless flexible air velocity microsensor was developed by using micro-electro-mechanical systems (MEMS) technology. Polyimide (PI) material was selected for the waterproof and oilproof requirements of the cold air duct environment of heating, ventilation, and air conditioning (HVAC) systems, and then a wireless flexible micro air velocity sensor was completed. To obtain real-time wireless measurements of the air velocity inside the cold air ducts of an HVAC system, and to create a measurements database, the deployment locations and quantity of micro air velocity sensors for the internal environment of the cold air ducts were planned. A field domain verification was performed to optimize the internal environment control of the cold air ducts of ventilation and air conditioning systems and to enhance the quality and reliability of process materials. This study realized real-time monitoring of velocity in the HVAC ducts of a chemical-fiber plant. A commercial velocity sensor (FS7.0.1L.195) was purchased and a micro-electro-mechanical systems (MEMS) approach was also used to develop a home-built micro air velocity sensor, to optimize the provision of the commercial sensors and our home-built micro air velocity sensor. Comparing the specifications of the two commercially available sensors with our home-built micro air velocity sensor, the results show that the home-built micro air velocity sensor has the advantages of fast response time, simultaneous sensing of three important physical quantities, and low cost.

## 1. Introduction

Heating, ventilation, and air conditioning (HVAC) refers to the process of discharging the air from an internal space or importing the outside air into an internal space, using mechanical equipment for ventilation, and importing the outside air to maintain the indoor air quality and reduce the moisture, peculiar smells, and contaminants in the air. However, if the outside humidity is high, extra energy is required to remove the moisture in the air [1]. The internal environment of the HVAC cold air duct, such as wind speed, temperature, and humidity, is closely related to the geographical environment where the plant is located. The introduction of a large amount of outside air (such as during a typhoon, heavy rain, etc.) will also indirectly affect the quality of the material. Therefore, in order to upgrade the quality and reliability of process materials, the internal environment of the cold air ducts is monitored, including air velocity, temperature, and humidity, and important physical quantities are provided for data analysis. Looking for the optimal internal environment control parameters is a very important research subject.

Zhang et al. [2] used ZnO/Al film to prepare a flexible gas flow sensor that had high sensitivity, quick responses, high reproducibility, and could maintain good performance in a bent pipe. To solve the difficulty in measuring the low air velocity of air-conditioning systems, Mu et al. [3] developed a novel damping torque air flow sensor. This sensor has a good linearity calibration curve in the damper angle range of 10°~50°. Shinoda et al. [4] studied the temperature measurement differences between nine wireless and two conventional wired commercial room temperature sensors, and compared the effects of indoor cooling systems, load, sensor types, and positions. They found that each sensor had a different sensitivity to radiation, and the influence of the radiation on the temperature measured by the wireless sensor was related to the difference between the black-ball temperature and air temperature. Chen et al. [5] proposed a back-off algorithm to reduce the humidity ratio error induced by the time constant of humidity and temperature sensors. Duy et al. [6] studied perovskite oxide SrTiO_3_ and successfully coated SrTiO_3_, which is difficult to calcine on a polyimide (PI) substrate by using the sol-gel process. This humidity sensor is flexible and sensitive to both capacitance and resistance. The resistance mode, with lower power consumption (approximately 0.2 μW), is applicable to long-term monitoring of relative humidity above 2%RH, while the capacitive mode, with high sensitivity, quick response/recovery (1–3 min), and low limit of detection (0.5%RH), can be used for calibration purposes. Asim et al. [7] reformed the existing HVAC system. Using the intelligent monitoring technique in HVAC systems can reduce the energy consumption of HVAC systems significantly. Ligęza [8] proposed a method and showed that two measured signals from an anemometer were proportional to the voltage and current of the sensors. The flow velocity was calculated by the dynamic model of sensors, the bandwidth of a measurement system was expanded, and the dynamic error was minimized. Ligęza [9] proposed a modified method for bandwidth measurement. The electrical test signals were directly used in the sensor in wireless mode by using transformer inductance coupling. Hardianto et al. [10] made a device that displayed real-time air velocity, wind direction, and temperature. The device used a Hall effect sensor and used a rotating magnet to generate pulse width modulation, and the sensor read air velocity. Ma et al. [11] studied the adsorption behaviors and sensing properties of oxygen for hydrogen and CO in different humidity conditions. The oxygen material adsorbed on the SnO_2_ surface varied with Pd loading, and Pd loading greatly enhanced the response of the sensors in a damp environment. Agrawal et al. [12] modified the shape of a micro-cantilever to enhance the sensitivity and reliability of an air velocity sensor based on micro-electro-mechanical systems (MEMS). Tang et al. [13] proposed a self-powered air velocity sensor based on a dual-mode TENG (triboelectric nanogenerator). The non-contact design was used to reduce wear and compensate for charge dissipation. Kou et al. [14] placed sensitive elements in a structure with a complementary split ring resonator (CSRR). The temperature, pressure, and humidity could be measured simultaneously. Fan et al. [15] proposed an iteration method based on the square wave disturbance response energy spectrum to achieve optimal frequency response and improve the frequency response of a constant temperature hot-wire anemometer. Duan et al. [16] used wind tunnel tests (WTT) and computational fluid dynamics (CFD) numerical simulation methods to study the environmental distinction of local wind at the Nei Lingding Island (NLDI) and Ping An Finance Center (PAFC) observation points in Shenzhen. The influence of the building units and instruments on the dead level at the top of NLDI and PAFC on the measurement result of the corresponding anemometer was analyzed. Ligęza [17] proposed the concept of a dynamic error calibration method for rotary anemometers and reported the results of the tests using this approach. The calibrations included accurately measuring the rotation speed of rotors and measuring the air velocity, and the instrument dynamics were considered. Ligęza [18] studied the influence of different types of windspeed sensors on measuring the windspeed profile in the standard region and described the region of the flow-blockage effect. The result shows that the flow blockage effect was minimized and the probability of standard correct placement was measured. Yi et al. [19] proposed a novel anemometer based on the inductance buckling effect. When the wind blows the flexible substrate, the surface inductor bends due to the air pressure. Therefore, the inductance reduction can be used as the measure of air velocity. Xu et al. [20] proposed a wind volume sensor system for detecting wind speed and wind direction simultaneously. Zhao et al. [21] demonstrated a novel wind power collector and self-powered wind power sensor based on a bionic triboelectric nanogenerator using a novel C-TENG structure made of an EPE framework. Lee et al. applied MEMS technology to exploit a flexible integrated microsensor, which can be inserted in the vanadium redox flow battery for real-time microscopic sensing and monitoring [22].

Heating, Ventilation, and Air Conditioning (HVAC) systems use mechanical devices to ventilate and bring in outside air to maintain indoor air quality. In order to improve the quality and reliability of process materials, it is important to monitor the internal environment of cold air ducts in real-time and find the best internal environmental control parameters. This study used micro-electro-mechanical systems (MEMS) technology to develop a micro air velocity sensor to be placed in HVAC vents [22].

## 2. Research Method

For wireless sensing of the air velocity inside the cold air ducts of HVAC systems, the deployment locations and quantity of different air velocity sensors for the internal environment of a cold air duct were planned. Commercial wireless sensors of different specifications were evaluated and purchased. Meanwhile, a wireless flexible micro air velocity sensor was developed to optimize the internal environment control of the cold air ducts of HVAC and to enhance the quality and reliability of process materials.

### 2.1. Design of Flexible Micro Air Velocity Sensor

A wireless flexible air velocity microsensor was developed in the current work by using micro-electro-mechanical systems (MEMS) technology. A PI material was selected for the waterproof and oilproof requirements of the cold air ducts environment of HVAC, and then the wireless flexible micro air velocity sensor was fabricated on this substrate.

The flexible air velocity microsensor is a hot-wire air velocity speed sensor, which is mainly composed of a heater. It uses the gas flow to take away the heat of the heater, so that the temperature of the heater drops, causing its resistance value to change. Thus, the air velocity speed of the gas can be calculated, as shown in Figure 1. It can be seen from Ohm’s law that, when the voltage is fixed, the resistance decreases and the current increases, so the calibration curve of wind speed and current change can be obtained, and then the wind speed of the gas can be obtained.

The sensing area of the flexible micro air velocity sensor is 400 µm × 400 µm. The line width and spacing of the sensor are 15 µm. The design and mask layouts are shown in Figure 2. The flowchart is shown in Figure 3. Surface micromachining technology was used in the manufacturing process, including coating (Figure 4a), exposure (Figure 4b), development, and lift-off (Figure 4c). The optical micrograph and stereogram of the finished flexible micro air velocity sensor are shown in Figure 4d,e.

In this study, an electron beam evaporation machine with fast film formation speed and high efficiency in Physical Vapor Deposition (PVD) was used to deposit the metal films onto the substrate. The advantages of this approach are fast film deposition and high single throughput. MEMS technology was applied to integrate air velocity sensors on a 50 µm-thick polyimide substrate. A 1500 Å thick gold layer was used as the sensing layer.

The microcontroller development board used in this study was the commonly used Arduino Uno Rev 3. Bluetooth can only receive signals; a Bluetooth transceiver must be connected to Raspberry Pi for conversion. With the Arduino microcontroller, a program can be written to receive the signals and convert them at the same time. With the selection of the Arduino, the readout program and circuit diagram of the sensor were developed and designed, thereby, finally, completing the monitoring module.

### 2.2. Comparisons between Commercial Air Velocity Sensors and the Home-Built Micro Air Velocity Sensor

The commercial sensors selected for the comparative study were the F660 and FS7.0.1L.195 commercial wireless air velocity sensors (Figure 5). A comparison between the performance of the two commercial air velocity sensors and the home-built microsensor is shown in Table 1. It was observed that the home-built micro air velocity sensor had faster responses and a lower cost than the two commercial air velocity sensors. The FS7.0.1L.195 air velocity sensor requires an external circuit; hence, the circuit conforming to the FS7.0.1L.195 specification was developed and is shown in Figure 6a. The packaging was designed according to the mechanism inside the cold air ducts of HVAC systems and is shown in Figure 6b. Finally, the split conductor was connected, and the signals were fed to the Arduino (the signal line was connected to the Arduino development board and linked to a computer during testing). The sensor was installed in a cold air duct at the test site. Since the cold air duct is a semi-closed space, the windspeed of the airflow is not exactly the same throughout the cold air duct. A sensor is connected to the Arduino for wireless data transfer by the port extension (Serial Port Expander).

## 3. Results and Discussion

### 3.1. Calibration of the Home-Built Wireless Flexible Micro Air Velocity Sensor

After the self-developed flexible micro air velocity sensor was completed, three home-built flexible micro air velocity sensors were soldered to a home-built circuit and connected to the signal transmission line of the Arduino development board. The data were read by the Arduino software, and testing and calibration were performed at different temperatures and air velocities of 1–10 m/s. Three readings are averaged in total. The original data was then normalized. The calibration is shown in Figure 7.

### 3.2. Calibration and Comparison of Home-Built Wireless Flexible Micro Air Velocity Sensor and FS7.0.1L.195 Commercial Air Velocity Sensor

The FS7.0.1L 195 commercial air velocity sensor was calibrated at a temperature of 25 °C and air velocities of 1–10 m/s, and compared with the calibration of the home-built flexible micro air velocity sensor, as shown in Figure 8. The difference between them was very small, indicating consistency.

## 4. Conclusions

This study consists of the specification and performance comparisons of two commercial air velocity sensors and a home-built flexible micro air velocity sensor. The process and performance optimization of the home-built flexible micro air velocity sensor and the integration of the home-built flexible micro air velocity sensor with a circuit was conducted. Furthermore, the home-built flexible micro air velocity sensor and commercial air velocity sensors were calibrated and compared in their performance. The findings show that the difference between the two kinds of devices was very small, indicating consistency.

## Figures and Tables

**Figure 1 micromachines-13-02198-f001:**
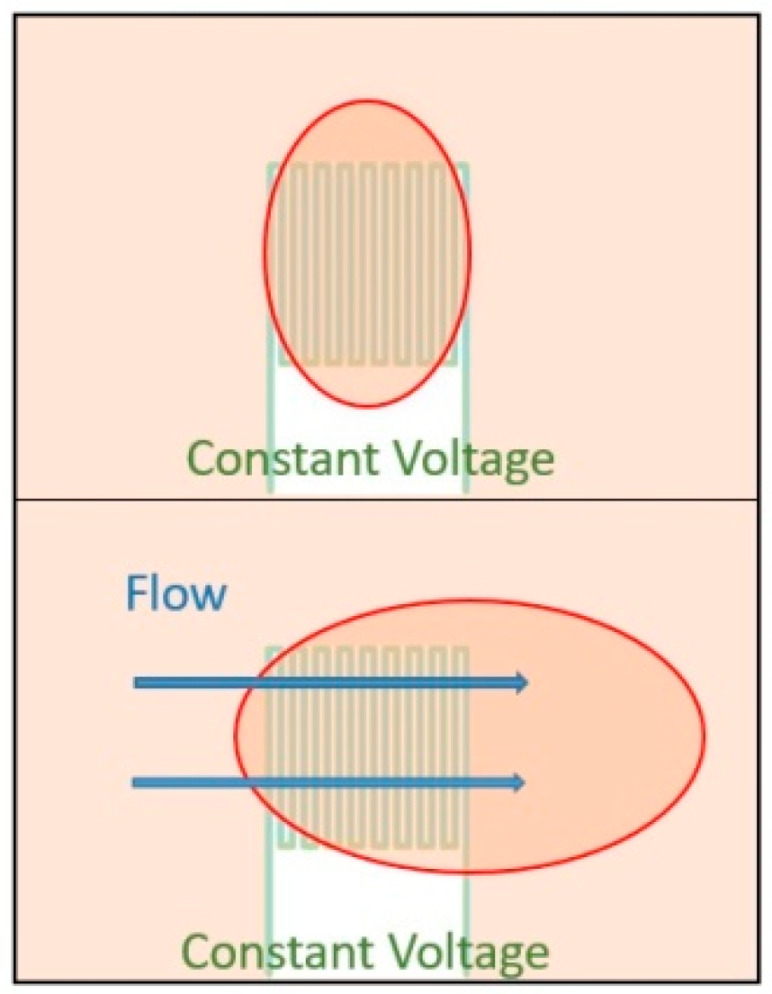
Schematic diagram of sensing with a flexible micro air velocity sensor.

**Figure 2 micromachines-13-02198-f002:**
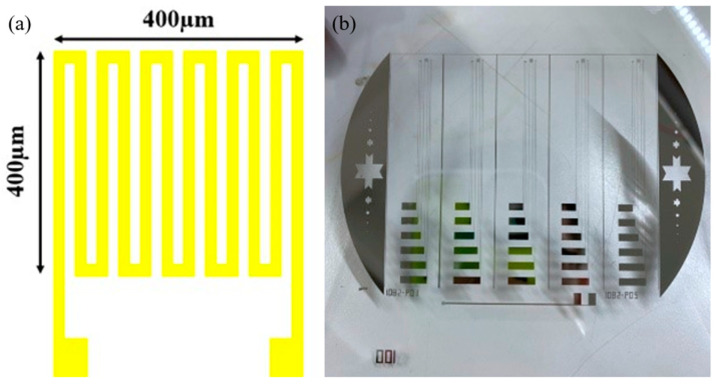
(**a**) Design diagram of the flexible micro air velocity sensor; (**b**) Mask layout of the flexible micro air velocity sensor.

**Figure 3 micromachines-13-02198-f003:**
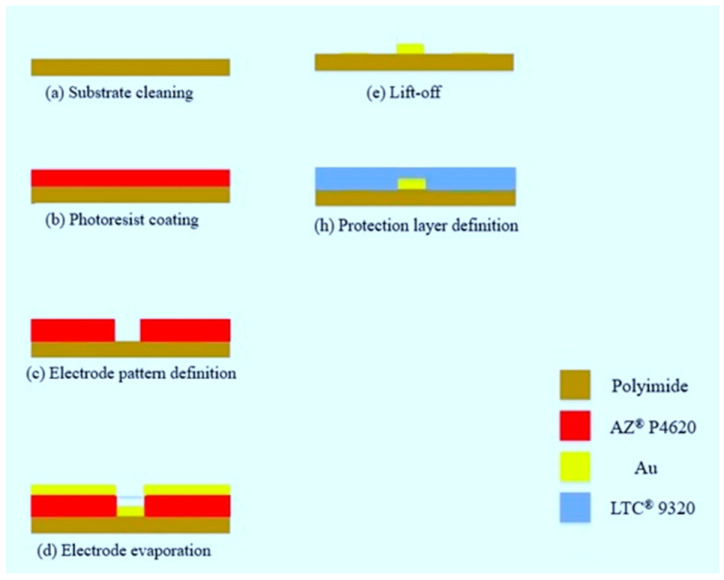
Fabrication process chart of the flexible micro air velocity sensor.

**Figure 4 micromachines-13-02198-f004:**
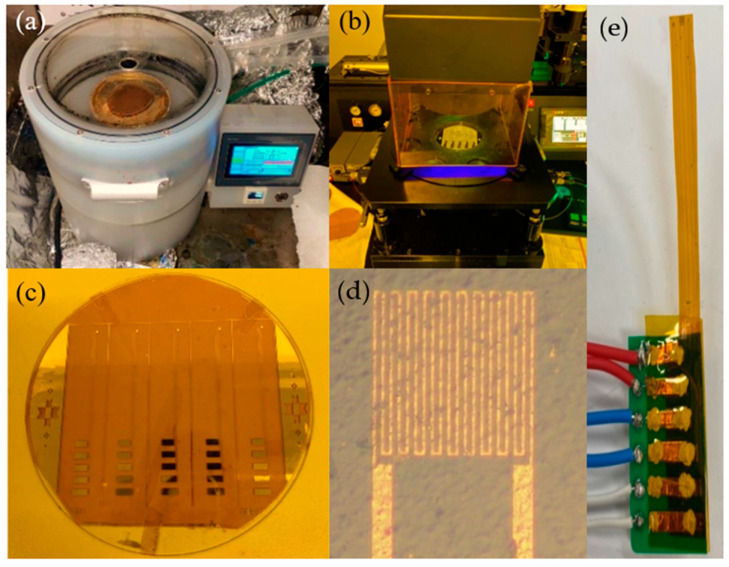
(**a**) Coating; (**b**) Exposure; (**c**) Lift-off; (**d**) Optical micrograph of the flexible micro air velocity sensor; (**e**) picture of the actual flexible micro air velocity sensor.

**Figure 5 micromachines-13-02198-f005:**
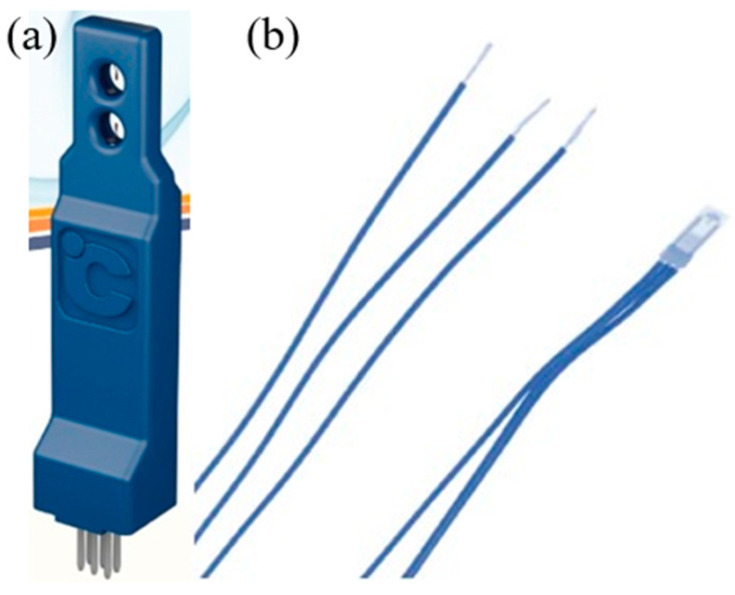
Commercial wireless air velocity sensors, (**a**) F660; and (**b**) FS7.0.1L.195.

**Figure 6 micromachines-13-02198-f006:**
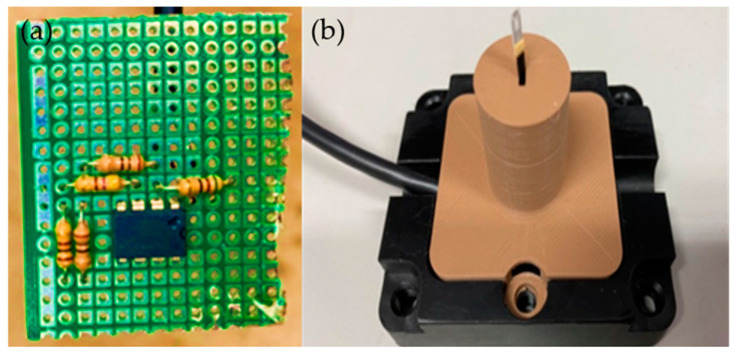
(**a**) External circuit board of the FS7.0.1L.195 wireless air velocity sensor; (**b**) photograph of self-developed packaging conforming to FS7.0.1L.195 dimensions.

**Figure 7 micromachines-13-02198-f007:**
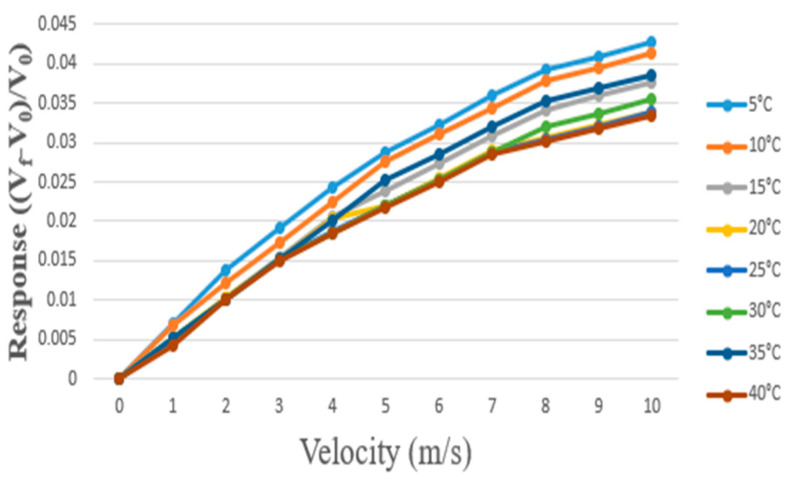
Calibration of home-built flexible micro air velocity sensor at different temperatures.

**Figure 8 micromachines-13-02198-f008:**
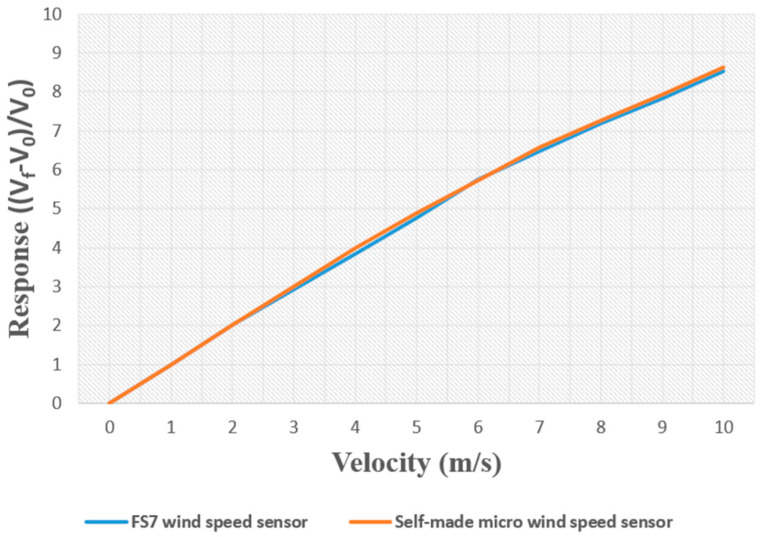
Calibration and comparisons of home-built flexible micro air velocity sensor and commercial air velocity sensor at 25 °C.

**Table 1 micromachines-13-02198-t001:** Specification and performance comparison of two commercial air velocity sensors and the home-built micro air velocity sensor.

Model	F660/F662 (Currently in Use)	FS7.0.0l.195	Home-Built Micro Sensor
Response time	400 ms	200 ms	1 ms
Operating measuring range	0.15–20 m/s	0–100 m/s	0.15–20 m/s
Operating temperature range	−5–60 °C	−20–150 °C	−20–250 °C
Output	Velocity, Temperature	Velocity	Velocity
Velocity accuracy	±5%	<3%	±3%
Price	NT4000	NT560	NT470
Length	39.1 mm	202 mm (including wire)	70 mm

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
