# Peer review of "A Micro Air Velocity Sensor for Measuring the Internal Environment of the Cold Air Ducts of Heating, Ventilation, and Air Conditioning Systems"

_micromachines, 2022, doi:10.3390/mi13122198_

Round 1

Reviewer 1 Report (New Reviewer)

The work entitled “A Micro Air Velocity Sensor for Measuring Internal Environment of Cold Air Pipe of Heating, Ventilation and Air Conditioning,” is a good experimental study of the air velocity sensor. Overall, in my opinion the work in not novel. I would suggest the following comments to the authors

1.     The abstract is not very well written. It does not give any information related to methodology, results, conclusion, and recommendations. In my opinion, generic lines should be removed and focused should be made on the strong statements and novelty of article. The authors should provide important quantity results from this research in the abstract and state how it is better than existing literature.

2.     The research gap should be provided based on existing literature very vividly.

3.     I encourage the authors to provide a table where the work is compared to existing literature based quotative values such as sensitivity, range, power consumption of the sensors. The authors compared their work with commercial sensors in table 1 but there is existing literature available on this topic and authors should include that in table 1.

4.     The quality of figures can be improved. Figure 9 and 10 look blur.

5.     Figures 4, 5, 7 and 8 should be combined in one figure

Author Response

Reviewer 2 Report (New Reviewer)

Journal Name: Micromachines (ISSN 2072-666X)

Manuscript ID: micromachines-2018069
Type of manuscript:
Communication

Title: A Micro Air Velocity Sensor for Measuring Internal Environment of Cold Air Pipe of Heating, Ventilation and Air Conditioning

Authors: Chi-Yuan Lee, Xin-Wen Wang, Chen-Kai Liu

In the reviewed communication, authors have designed and developed an air velocity wireless flexible microsensor by using MEMS technology. In order to make it oil-proof and water-proof specifically for HVAC application purposes, they have used polyimide (PI) material. The performance of the developed sensor is compared with two commercial sensors (F660 and FS7.0.1L.195). The following suggestions may be incorporated in the revised manuscript:

·       1# In the last paragraph of the introduction, the existing literature gap and novelty may be written.

·        2#   In section 2.1, the design aspects of the microsensor have been highlighted. On page 3 lines 107-113, the working principle of the sensor is elaborated.

·         3# In section 2.1 lines 116 and 120, the value of ‘n’ (i.e., the correlation coefficient of U and Q) is chosen to be zero. Did the authors have calculated the value of ‘n’ by their own experimental investigation or taken from other sources?

·          4On page 3 lines 123 to 133, the structural details of the microsensor are written and fabrication methodologies are highlighted. It is suggested to mention the fabrication challenges and necessary remedies.

·         5# On page 6 Table 1, the specifications of the self-made microsensor are compared against two commercial microsensors (F660 and FS7.0.1L.195). It is noticed that the accuracy of the self-made sensor is ±3% and quite comparable against commercial sensor FS7.0.1L.195. But, its size is lower than FS7.0.1L.195, which will improve its reliability for many practical applications. The price of the self-made sensor is lower than the price of both commercial sensors. The repeatability and typical working conditions of the self-made sensor may be compared against the commercial sensors in Table 1.

·         6 # In section 3 line 173, it is mentioned that the sensor is calibrated for different temperatures and calibration results are plotted in Fig. 9. Did the working temperature range lie between 15°C to 40°C? Will the sensor work efficiently beyond this range? It is suggested to test its performance beyond this range of temperature and report its accuracy and repeatability. How the sensor will behave in cryogenic temperature limits typically i.e., below 120 K? What is the maximum and minimum working temperature limit of the sensor? Will the sensor be able to measure oscillating air velocity? Will the sensor be able to measure the velocity of other gases (like Helium, Nitrogen, Argon, Oxygen etc.) except air?

·          7# The confidence level of the experimental investigation needs to be addressed.

·         8# The influence of remaining atmospheric conditions on the performance of the sensor may be addressed shortly. 

Overall, the manuscript is well-written and grammatically correct. Calibration graphs of self-made sensor are also matching with commercial sensors. It may be accepted for publication after addressing the limitations mentioned above.

*********

Author Response

This manuscript is a resubmission of an earlier submission. The following is a list of the peer review reports and author responses from that submission.

Round 1

Reviewer 1 Report

English language and presentation style need major revision. Though authors have responded to most of comments , presentation style has not improved . It is major draw back of the manuscript .

For example line 129-133 ( and a few more)  about electron beam evaporation (which are otherwise not required ) can be written in more lucid manner. 

 Lines 13-17 mention the the 'deployment  locations and quantity of micro air velocity sensors for the internal environment monitoring'.  It is not discussed in the paper.

Reviewer 2 Report

In the revised version the authors only added some words, combine several figures, but did not do anything on experiments. It does not meet the requirements for publication.